# Identification of Fucosylated SERPINA1 as a Novel Plasma Marker for Pancreatic Cancer Using Lectin Affinity Capture Coupled with iTRAQ-Based Quantitative Glycoproteomics

**DOI:** 10.3390/ijms22116079

**Published:** 2021-06-04

**Authors:** Chia-Chun Wu, Yu-Ting Lu, Ta-Sen Yeh, Yun-Hsin Chan, Srinivas Dash, Jau-Song Yu

**Affiliations:** 1Molecular Medicine Research Center, Chang Gung University, Taoyuan 33302, Taiwan; wuchiachun9@gmail.com (C.-C.W.); judy1143039@gmail.com (Y.-T.L.); 2Department of General Surgery, Chang Gung Memorial Hospital, Linkou 33305, Taiwan; tsy471027@cgmh.org.tw (T.-S.Y.); fm423344@gmail.com (Y.-H.C.); 3College of Medicine, Chang Gung University, Taoyuan 33302, Taiwan; 4Graduate Institute of Biomedical Sciences, College of Medicine, Chang Gung University, Taoyuan 33302, Taiwan; srinivasdash26@gmail.com; 5Liver Research Center, Chang Gung Memorial Hospital, Linkou 33305, Taiwan; 6Research Center for Food and Cosmetic Safety, College of Human Ecology, Chang Gung University of Science and Technology, Taoyuan 33302, Taiwan

**Keywords:** pancreatic cancer, plasma, glycobiomarker, AAL, iTRAQ-based quantitative proteome, reverse lectin-based ELISA, fucosylated SERPINA1

## Abstract

Pancreatic cancer (PC) is an aggressive cancer with a high mortality rate, necessitating the development of effective diagnostic, prognostic and predictive biomarkers for disease management. Aberrantly fucosylated proteins in PC are considered a valuable resource of clinically useful biomarkers. The main objective of the present study was to identify novel plasma glycobiomarkers of PC using the iTRAQ quantitative proteomics approach coupled with *Aleuria aurantia* lectin (AAL)-based glycopeptide enrichment and isotope-coded glycosylation site-specific tagging, with a view to analyzing the glycoproteome profiles of plasma samples from patients with non-metastatic and metastatic PC and gallstones (GS). As a result, 22 glycopeptides with significantly elevated levels in plasma samples of PC were identified. Fucosylated SERPINA1 (fuco-SERPINA1) was selected for further validation in 121 plasma samples (50 GS and 71 PC) using an AAL-based reverse lectin ELISA technique developed in-house. Our analyses revealed significantly higher plasma levels of fuco-SERPINA1 in PC than GS subjects (310.7 ng/mL v.s. 153.6 ng/mL, *p* = 0.0114). Elevated fuco-SERPINA1 levels were associated with higher TNM stage (*p* = 0.024) and poorer prognosis for overall survival (log-rank test, *p* = 0.0083). The increased plasma fuco-SERPINA1 levels support the utility of this protein as a novel prognosticator for PC.

## 1. Introduction

Pancreatic cancer (PC) is the most lethal malignant disease associated with a high mortality rate. The 5-year survival rates of PC are reported as ~8% and 9% in the United States [1] and European Union [2], respectively. In 2016, PC was ranked the eighth leading cause of cancer-related mortality in Taiwan. Since the majority of early-stage (stage I or II) PC cases are asymptomatic, ~80% patients present at later stages (stage III or IV) of disease progression with metastatic spread and unresectable tumors at the time of diagnosis [3,4]. For patients with locally resectable non-metastatic disease, surgical resection followed by adjuvant chemotherapy is the main treatment modality. Chemotherapy administered to patients with advanced disease is frequently associated with treatment resistance and unfavorable side effects. Effective management of PC thus remains a major challenge [5]. Reliable diagnostic, prognostic and predictive biomarkers of PC for better patient stratification and guidance of therapy choices remain an urgent medical requirement. Among the currently available tumor biomarkers, CA19-9 is commonly used to monitor PC. However, CA19-9 is not an effective biomarker for early detection in view of its false-positive results in chronic pancreatitis, biliary tract inflammation and cancers in other organs, including stomach, colon, ovary, uterus and liver [6,7].

Protein glycosylation, the most abundant posttranslational modification, plays fundamental roles in protein function. Aberrant protein glycosylation is known to regulate numerous disease processes including malignant transformation [8]. For example, previous studies have reported significant involvement of aberrant protein fucosylation in tumorigenesis of PC [9,10]. To facilitate systematic identification of glycoproteins involved in the regulation of normal biological functions and/or disease processes, the lectin-based approach has been employed to enrich glycopeptides for proteomic analysis [11,12]. Glycoproteomic analysis of *N*-linked glycopeptides using lectin-affinity capture coupled with isotope-coded glycosylation site-specific tagging (IGOT) and isobaric tag for relative and absolute quantitation (iTRAQ) labeling has also been developed to quantify and identify glycopeptides with accurate glycosylated sites [11,12]. In 2014, Nie et al. [13] used a microarray of 16 lectins to identify *Aleuria aurantia* lectin (AAL), which recognizes terminal α-linked fucose, as a useful tool to distinguish PC. The group further applied AAL-affinity capture coupled with TMT labeling and label-free approaches to identify three serum proteins (α-1 antichymotrypsin, thrombospondin-1 and haptoglobin) as a potential marker panel for PC detection [13].

Following the initial discovery of dysregulated glycoproteins in biological samples, researchers often need to develop methods to quantify glycosylation changes of specific target glycoproteins in a large number of samples, such as serum/plasma, in a high-throughput manner. For example, lectin-based antibody microarrays and lectin-based immunosorbent assays (lectin-ELISA) have been developed, both of which are based on coating antibodies to slides or 96-well plates followed by capture of antigens in samples and subsequent detection of captured antigens by lectin [14,15,16]. However, glycans on the antibodies used for coating to slides or plates may interfere with the assays, and further experimental designs to diminish this type of interference are required [17,18]. To circumvent this issue, a reverse lectin-based ELISA system whereby specific lectins are coated to slides or plates followed by capture of glycoproteins in samples and their detection with specific antibodies has been applied successfully to quantify different glycoproteins in serum/plasma samples [19,20].

In the present study, AAL was used as a glycopeptide enrichment tool for identifying novel biomarkers from abundant protein-depleted plasma samples collected from PC patients and subjects with gallstones (GS) by application of IGOT coupled with MS analysis. Numerous plasma glycopeptides upregulated in PC patients compared to controls and predicted to harbor core or antennary fucose were identified. Among them, fucosylated SERPINA1 (fuco-SERPINA1) displaying the highest fold change between the metastatic PC and GS groups was selected for further verification. Furthermore, a reverse lectin-based ELISA assay was developed in-house to evaluate the biomarker potential of fuco-SERPINA1.

## 2. Results

### 2.1. Study Population and Experimental Design

We recruited 30 subjects (10 metastasis-free PC patients (M0), 10 PC patients with distant metastasis (M1) and 10 subjects with gallstones (GS)) for the discovery experiment and an additional 91 subjects (40 GS and 51 PC) for verification experiments. The clinicopathological characteristics of the enrolled subjects are shown in Table 1. Plasma samples were collected from all subjects for measurement of proteins and glycoproteins. To establish useful plasma glycobiomarkers for PC, three groups of pooled plasma samples from GS, M0 and M1 groups (10 cases per group) were subjected to depletion of the top 14 high-abundance proteins followed by iTRAQ labeling (with three plex 114, 115 and 116) and 2D-LC-MS/MS analysis for systemic comparison of differential expression of proteins and glycoproteins. For glycoproteomic analysis, *Aleuria aurantia* lectin (AAL), identified as an ideal lectin for discrimination of PC from normal and other pancreatitis cases [13], was applied to enrich glycopeptides and *N*-glycosidase (PNGase F) used to eliminate glycan from asparagine (Asn) in H_2_^18^O-containing solution for transferring Asn to aspartic acid (Asp) with ^18^O labeling to enhance the accuracy of identifying glycopeptides in MS analysis. ^18^O-labeled glycopeptides containing fucose and upregulated in PC were selected as targets for verification. Glyco-SERPINA1, one of the candidate targets in plasma samples of PC and GS, was further validated using reverse AAL-based ELISA. The workflow of our study design is presented in Figure 1.

### 2.2. iTRAQ-Based Quantitative Glycoproteomics Coupled with Glycopeptide Enrichment via AAL-Affinity Capture Technique for Identification of Plasma Glycobiomarkers

Using the strategy shown in Figure 1, a total of 1707 proteins (10,302 peptides) and 2160 proteins (10,572 peptides) were respectively identified in the plasma proteome and glycoproteome profiles of the three groups (GS, M0 and M1) in the discovery experiment (Table 2). Detailed identification and quantification of peptides/proteins and glycopeptides/glycoproteins are presented in Appendix A. Quantitative proteome profiling facilitated the quantification of 1489 and 1472 proteins in M0/GS and M1/GS groups, respectively. For glycoproteome analysis, 281 and 267 ^18^O-labeled peptides containing the N to D modification (N to D_^18^O peptides) were quantified in M0/GS and M1/GS groups, corresponding to 133 and 130 N to D_^18^O proteins, respectively. The ratio of N to D_^18^O peptides to total identified peptides was 3.01–3.32%. Among the N to D_^18^O peptides identified, ~87% (244 of 281 in the M0/GS group, 232 of 267 in M1/GS group) possessed the consensus *N*-glycosylation motif (NXS/T/C, X represents any amino acid except proline) (Table 2).

### 2.3. Selection of Candidate Plasma Glycomarkers of PC through Integrating Expression of Glycopeptides and Their Glycan Compositions

The flowchart of candidate glycopeptide selection is depicted in Figure 2. To identify candidate glycopeptides, we first selected 441 peptides displaying upregulation (≥mean + S.D. 1.36 for M0/GS group, 2.01 for M1/GS group) in both M0 and M1 compared to the GS group and further identified 27 peptides with N to D_^18^O modification. After removal of redundant peptides, 22 peptides derived from 19 proteins were selected as candidates and their protein ratios (determined via quantitative proteome analysis) were applied to normalize the glycopeptide ratio for measuring changes in their glycosylation levels (Table 3). The representative MS/MS spectra of the 22 peptides are shown in Appendix A. In total, 15 of the 22 target peptides were identified with glycan modifications, among which 12 were detected with fucose decoration (Table 3 and Appendix A).

Among the 12 fucose-containing peptide candidates, SERPINA1 was selected for further validation since the glycopeptide (YLGNATAIFFLPDEGK) displayed significantly elevated levels in PC (including both M0 and M1 groups) versus GS samples, with the highest fold change between the M1 and GS groups (5.22) among the fucose-containing peptides. Figure 3 depicts the in-depth resolved glycan structure of SERPINA1 (YLGNATAIFFLPDEGK) based on analysis using Byonic software (Figure 3a,b) as well as the indicated peaks with fucosylated glycan in the collision-induced dissociation fragment spectrum (Figure 3c).

**Table 3 ijms-22-06079-t003:** List of 22 glycopeptides showing significantly elevated levels in plasma samples of PC patients compared to GS controls.

Gene Name	Protein Name	Sequence [N(n) to D_^18^O] *^a^*	Modified Site	M0/GS			M1/GS			Glycan Occupancy	Fucosylated Glycan
				Gp *^b^*	*p ^c^*	Gp/*p ^d^*	Gp	*p*	Gp/*p*		
APOH	Beta-2-glycoprotein 1	R.VYKPSAGnNSLYR.D	N162	1.97	1.18	1.66	3.21	1.70	1.89	V	V
ATRN	Attractin	R.nHSCSEGQISIFR.Y	N731	1.89	1.20	1.57	2.35	1.54	1.53	V	V
AZGP1	Zinc-alpha-2-glycoprotein	R.FGCEIEnNR.S	N127	1.36	1.20	1.14	2.28	2.00	1.14		
CD14	Monocyte differentiation antigen CD14	R.nVSWATGR.S	N151	1.88	1.38	1.37	2.65	1.60	1.66	V	V
CD163	Scavenger receptor cysteine-rich type 1 protein M130	K.APGWAnSSAGSGR.I	N105	2.13	1.78	1.20	3.00	2.46	1.22	V	V
CD163	Scavenger receptor cysteine-rich type 1 protein M130	K.EDAAVnCTDISVQK.T	N1027	1.64	1.78	0.92	2.41	2.46	0.98		
CTSD	Cathepsin D	K.GSLSYLnVTR.K	N263	3.15	2.61	1.20	2.51	2.23	1.12	V	
ICAM1	Intercellular adhesion molecule 1	R.LNPTVTYGnDSFSAK.A	N267	3.05	2.48	1.23	3.16	2.42	1.30	V	V
IL18BP	Interleukin-18-binding protein	K.ALVLEQLTPALHSTnFSCVLVDPEQVVQR.H	N147	2.87	1.99	1.44	4.12	1.95	2.11		
IL6ST	Interleukin-6 receptor subunit beta	K.EQYTIInR.T	N83	1.47	1.39	1.06	2.10	1.87	1.12	V	V
LEPR	Leptin receptor	K.YSEnSTTVIR.E	N276	2.41	1.64	1.47	2.28	2.00	1.14		
LRG1	Leucine-rich alpha-2-glycoprotein	K.MFSQnDTR.C	N325	1.61	1.30	1.24	3.39	2.32	1.46	V	
LRG1	Leucine-rich alpha-2-glycoprotein	R.KLPPGLLAnFTLLR.T	N186	1.57	1.30	1.21	2.96	2.32	1.28	V	V
LRG1	Leucine-rich alpha-2-glycoprotein	K.LPPGLLAnFTLLR.T	N186	1.43	1.30	1.10	2.15	2.32	0.93	V	V
LUM	Lumican	R.LSHNELADSGIPGnSFNVSSLVELDLSYNK.L	N249	2.51	1.17	2.14	4.04	1.74	2.32		
MMRN1	Multimerin-1	K.FNPGAESVVLSnSTLK.F	N136	2.61	0.68	3.82	3.32	0.84	3.93	V	V
ORM1	Alpha-1-acid glycoprotein 1	R.QDQCIYnTTYLNVQR.E	N93	2.34	2.02	1.16	5.39	4.73	1.14		
OSMR	Oncostatin-M-specific receptor subunit beta	R.SVNILFnLTHR.V	N326	1.54	1.31	1.18	2.14	1.43	1.49	V	V
PRNP	Major prion protein	K.GEnFTETDVK.M	N197	1.42	N/A *^e^*	N/A	2.70	N/A	N/A		
SERPINA1	Alpha-1-antitrypsin	K.YLGnATAIFFLPDEGK.L	N271	1.87	1.57	1.19	5.22	3.74	1.40	V	V
SERPINC1	Antithrombin-III	K.SLTFnETYQDISELVYGAK.L	N187	1.95	1.18	1.66	2.58	1.46	1.77	V	V
VASN	Vasorin	R.LHEITnETFR.G	N117	1.46	1.49	0.98	2.21	1.71	1.29	V	

*^a^* Sites with *N*-glycosylation of N to D_^18^O modification are presented in bold and lower case. *^b^* Gp, ratio of glycopeptide obtained from glycoproteome analysis. *^c^ p*, ratio of protein obtained from proteome analysis. *^d^* Gp/*p*, level changes of glycosylation. *^e^* N/A, protein not identified in proteome analysis.

### 2.4. Removal of Glycans from SERPINA1 Protein by PNGase F Blocks Its AAL Lectin Binding Activity

To examine potential glycosylation on SERPINA1, we treated plasma proteins obtained from PC patients and a commercially available recombinant human SERPINA1 protein (derived from a mouse myeloma cell line NS0) with PNGase F to eliminate glycan. SERPINA1 was subsequently detected via Western and AAL lectin blots. As shown in Figure 4, Western blot using an anti-SERPINA1 antibody led to the detection of a major 60 kDa protein band in both PC plasma and recombinant human SERPINA1 protein samples prior to PNGase F treatment. Another protein band with lower apparent molecular weight (~57 kDa) emerged clearly after PNGase F treatment. AAL lectin blot analysis showed strong lectin binding signal of the 60 kDa recombinant human SERPINA1, which was significantly diminished after PNGase F treatment (Figure 4b, middle panel). Importantly, the ~57 kDa SERPINA1 protein band completely lacked the AAL binding signal. The results collectively suggest that SERPINA1 is modified via glycosylation and its glycan structures are accessible for recognition by AAL. 

### 2.5. Development of AAL-Based Reverse Lectin ELISA for Measuring Glycosylated SERPINA1 Levels

To measure the levels of glycosylated SERPINA1 in individual samples, we developed an AAL-based reverse lectin ELISA technique and evaluated its specificity for glycosylated SERPINA1. Recombinant SERPINA1 protein harboring AAL-specific glycan was used as a standard. Dynamic range of detecting glycosylated SERPINA1 was determined from 1.563 to 800 ng/mL based on the standard curve generated using linear dilutions of recombinant SERPINA1 proteins (Appendix A). To evaluate the specificity of AAL binding to glycosylated SERPINA1 via recognition of fucose in this system, we initially examined the effect of PNGase F treatment on the AAL binding capability of recombinant SERPINA1. PNGase F-catalyzed deglycosylation of SERPINA1 completely abolished recognition of AAL (Figure 5a). Next, we investigated the effect of addition of different sugar types in reverse lectin ELISA. The ELISA signal was suppressed by L-fucose but not lactose in a dose-dependent manner (Figure 5b). The collective results suggest that fucose is a pivotal component of glycan for AAL binding to glycosylated SERPINA1 and validate the effective application of our newly developed AAL-based reverse lectin ELISA assay to detect fuco-SERPINA1 in clinical plasma samples.

### 2.6. Changes in SERPINA1 and Fuco-SERPINA1 Levels in Individual Plasma Samples

The AAL-based reverse lectin ELISA system was applied to determine the levels of fuco-SERPINA1 in plasma samples of 121 subjects (50 GS and 71 PC) enrolled in this study. Levels of fuco-SERPINA1 were significantly higher in PC than GS patients (310.7 ng/mL v.s. 153.6 ng/mL, *p* = 0.0114) (Figure 6a). Notably, levels of fuco-SERPINA1 were also significantly higher in PC patients with distant metastasis (M1) than the metastasis-free (M0) PC group (M0: 228.6 ng/mL; M1: 361.1 ng/mL, *p* = 0.043) (Figure 6b). We additionally measured the levels of SERPINA1 protein in the same sample set using a commercial ELISA kit with a detection range of 7.813 to 8000 µg/mL (Appendix A). The results consistently revealed significant elevation of SERPINA1 protein in PC compared to GS (139.9 µg/mL v.s. 106.1 µg/mL, *p* < 0.0001), but the extent of change (1.32-fold) was lower than that of fuco-SERPINA1 (2.02-fold) (Figure 6c). Moreover, patients with distant metastasis of PC had slightly higher SERPINA1 levels compared to metastasis-free patients (129.7 µg/mL v.s. 146.1 µg/mL, *p* = 0.017) (Figure 6d). Levels of CA19-9 in the 50 GS and 71 PC subjects were additionally determined. The results showed dramatic elevation of CA19-9 levels in PC patients relative to GS subjects (Figure 6e,f). Taken together, these results confirmed our findings from the quantitative glycoproteomics study, demonstrating that (i) both plasma SERPINA1 and fuco-SERPINA1 levels are significantly elevated in PC compared to GS patients, and (ii) the observed increase in plasma fuco-SERPINA1 levels is mainly due to significant elevation in cases of PC with distant metastasis.

### 2.7. Associations of Plasma Levels of Fuco-SERPINA1, SERPINA1 and CA19-9 with Clinicopathological Characteristics of PC Patients

Next, we explored the potential association of plasma levels of fuco-SERPINA1, SERPINA1 and CA19-9 with different clinicopathological characteristics (gender, age, TNM stage, tumor stage, lymph node metastasis and distant metastasis) of enrolled PC patients (Table 4 and Appendix A). The results showed that (i) all three measurements (fuco-SERPINA1, SERPINA1 and CA19-9) are not significantly associated with gender or age, (ii) higher levels of fuco-SERPINA1 are significantly correlated with higher TNM stage (*p* = 0.024) and distant metastatic PC (M1) (*p* = 0.043), (iii) SERPINA1 protein levels are significantly correlated with distant metastasis at diagnosis (*p* = 0.017), and (iv) plasma levels of CA19-9 do not show a significant correlation with the clinical characteristics, although subjects with distant metastatic PC (M1) at diagnosis tend to have higher CA19-9 levels (*p* = 0.097).

### 2.8. Receiver Operating Characteristic (ROC) Curve Analysis of Fuco-SERPINA1, SERPINA1 and CA19-9

The efficacy of fuco-SERPINA1, SERPINA1 or CA19-9 for discriminating between PC patients and GS controls was assessed via ROC curve analysis. AUC (Area Under ROC Curve) values of fuco-SERPINA1, SERPINA1 and CA19-9 were determined as 0.652, 0.836 and 0.914, respectively (Figure 7a). The plasma levels of CA19-9 displayed outstanding performance in discriminating PC patients from GS controls. Discriminatory power was also high for SERPINA1 protein but poor for fuco-SERPINA1. Combination of fuco-SERPINA1 with SERPINA1 protein or CA19-9 did not enhance the discriminating power of CA19-9 or SERPINA1 protein alone. On the other hand, the discriminatory power of the combination of SERPINA1 protein and CA19-9 was greater than that of either marker alone (AUC = 0.956) (Figure 7b).

### 2.9. Association of Overall Survival (OS) with Fuco-SERPINA1, SERPINA1 and CA19-9

To investigate the association of patient survival with plasma levels of fuco-SERPINA1 and SERPINA1, their median levels were used as cut-off values (98.7 ng/mL for fuco-SERPINA1; 147.1 µg/mL for SERPINA1 protein). Patients were stratified into high- and low-level groups and Kaplan–Meier plots generated to estimate OS rates. The cut-off value for CA19-9 was set at 37 U/mL. OS rates were determined in 70 of the 71 PC patients. PC patients with high fuco-SERPINA1 levels (*n* = 35) had significantly lower survival rate than those with low fuco-SERPINA1 levels (*n* = 35; *p* = 0.0083). Similarly, higher SERPINA1 protein and CA19-9 levels were respectively associated with poorer survival rates (*p* < 0.0001; *p* = 0.0109) (Figure 8). Our results suggest that plasma levels of both fuco-SERPINA1 and SERPINA1 protein may serve as effective novel prognosticators of PC.

## 3. Discussion

Abnormal glycosylation is considered a hallmark associated with cancer [8]. Specific glycoproteins incorporating aberrant glycans have been uncovered with significantly higher specificity for cancers than the proteins themselves, such as glycosylated alpha-fetoprotein (AFP), which serves as a more reliable marker for hepatocellular carcinoma (HCC) than AFP protein [21,22]. Several studies have reported abnormal fucosylation in colorectal and prostate cancer types [23,24] and increase in haptoglobin decorated with glycan of core fucosylation (α-1-6 linked fucose on *N*-acetylglucosamine at the reducing end) in serum samples of PC patients [10,25].

Previously, a lectin-based quantitative proteomics approach was applied to identify potential serum glycomarkers for PC. Six candidate proteins were verified using ELISA (alpha-1-antichymotrypsin (AACT), alpha-1-antitrypsin (A1AT), leucine-rich alpha-2-glycoprotein (LRG), thrombospondin-1 (THBS1) and haptoglobin (HPT)) or AAL lectin-ELISA (HPT and lumican (LUM)) in serum samples from 34 PC patients and 142 non-PC controls, which yielded a three-marker panel (AACT, THBS1 and HPT) with higher diagnostic potency for PC than the single biomarker CA19-9 [13]. Interestingly, the group observed elevated serum levels of α-1-antitrypsin (A1AT, also known as SERPINA1) in PC patients compared to non-PC controls. In the present study, we adopted a different experimental design by incorporating isotope-coded glycosylation-site-specific tagging (IGOT) into the lectin-based quantitative proteomics approach [26], which facilitated the identification of 22 glycopeptides with definite glycosylation sites that were significantly elevated in plasma samples of PC compared to non-PC subjects (Table 3). The majority of glycosylation sites, except Asn 325, Asn 186 and Asn 186 for LRG1, Asn 136 for MMRN1 and Asn 117 for VASN, have been reported in UniPep, a database of human *N*-linked glycosites (http://www.unipep.org accessed on 31 March 2021) [27], validating the robustness of our assay platform for accurate identification of glycosylation sites. Moreover, analysis using Byonic software led to successful elucidation of the glycan structures of 12 fucose-containing glycopeptides, including ^268^YLGNATAIFFLPDEGK^283^ derived from SERPINA1. To our knowledge, the majority of differentially expressed glycopeptides identified in our experiments have not been reported in earlier PC studies.

SERPINA1 (alpha-1-antitrypsin) is an inducible gene mainly expressed in hepatocytes, monocytes and macrophages. It encodes for AAT, a serine protease inhibitor mainly synthesized by the liver, and a highly expressed glycoprotein released into the bloodstream [28]. It acts as an inhibitor of neutrophil elastase, trypsin, chymotrypsin, thrombin, plasmin and cathepsin G. Deficiency of this protein mainly causes chronic obstructive pulmonary disease [29,30]. Several studies have reported an increase in serum SERPINA1 protein levels in the pregnancy, inflammatory response and different malignancy types derived from lung, liver, stomach, colon, prostate and pancreas [31,32,33,34,35,36,37]. However, little is known about glycosylation in PC although an increased level of serum fucosylated SERPINA1 has been reported in patients with lung cancer or hepatocellular carcinoma [38,39]. Previous LC-MS/MS analyses have led to the identification of three *N*-glycosylation sites of standard serum protein SERPINA1: Asn 70 (decorated with di-antennary glycan without fucose), Asn 107 (decorated with core and peripheral fucose linked to di-, tri- and tetra-antennary glycans) and Asn 271 (decorated with core and peripheral fucose linked to di- and tri-antennary glycans) [40]. Among the three sites, glycosylation of SERPINA1 at Asn 271 was significantly higher in plasma samples of PC patients (especially for the distant metastatic PC subgroup) (Table 3), with the deduced structure containing di-antennary glycans displaying both core and antennary fucosylation (Figure 3). Importantly, we successfully developed sensitive AAL-based reverse lectin ELISA in this study for measurement of fuco-SERPINA1 levels in plasma specimens, and reported for the first time that both plasma fuco-SERPINA1 and SERPINA1 protein levels are significantly elevated in patients with distant metastatic PC (Figure 6) in association with poor prognosis in OS (Figure 8). However, the PC sample used in this study was relatively small and the utility of fuco-SERPINA1 and SERPINA1 proteins as novel prognosticators for PC should be examined with larger sample numbers in the future.

Except SERPINA1, 10 targets, including beta-2-glycoprotein 1 (APOH), attractin (ATRN), monocyte differentiation antigen CD14 (CD14), scavenger receptor cysteine-rich type 1 protein M130 (CD163), intercellular adhesion molecule 1 (ICAM1), interleukin-6 receptor subunit beta (IL6ST), leucine-rich alpha-2-glycoprotein (LRG1), multimerin-1 (MMRN1), oncostatin-M-specific receptor subunit beta (OSMR) and antithrombin-III (SERPINC1), were also identified with fucosylation on their glycans. Among them, five targets (APOH, CD14, ICAM1, LRG1 and SERPINC1) were reported to be associated with other cancers or prognostic significance. For example, fucosylated APOH, CD14 and ICAM1 were detected with a high level in plasma of patients with human hepatocellular carcinoma (HCC) and fucosylated ICAM1 may represent a good prognostic marker for HCC [41,42,43]. In addition, fucosylated ICAM1 was identified as a potential biomarker for distinguishing Hodgkin’s lymphoma from other lymphocytic cancers [44]. Higher serum levels of SERPINC1 with sialylation and fucosylation in PC patients as compared to normal or chronic pancreatitis subjects has been reported [45]. The plasma levels of core-fucosylated LRG1 were found to be elevated in oral cancer patients relative to normal cases; moreover, the serum levels of LRG1 with fucosylated triantennary *N*-glycan was identified as a new marker to distinguish colorectal cancer patients from healthy, with a sensitivity and specificity exceeding CA19-9 [46,47].

In addition, the plasma levels of five glycopeptides derived from fructose-bisphosphate aldolase A (ALDOA), HPT, hemopexin (HPX), SERPINA1 and 14-3-3 protein theta (YWHAQ) were found to be significantly elevated in the M1 group compared with the M0 group (≥mean + S.D. 2.57) (Appendix A). Among them, two proteins (HPT and HPX) in their sialylated and fucosylated forms have been reported as potential biomarkers for PC [13,45]. Notably, studies have unraveled core and peripheral fucose linked to di- and tri-antennary glycans as the glycoconjugate structures of HPT, and patients with late-stage PC showed significantly higher serum fucosylated HPT (fuco-HPT) levels than those of early-stage PC patients and healthy subjects [10,48]. Therefore, it may be worth to examine the potential of combining fuco-SERPINA1 and fuco-HPT as biomarker panel to improve the detection of metastatic PC.

We used an AAL affinity column to enrich iTRAQ-labeled glycopeptides from plasma samples. This peptide level-based protocol is more efficient for enrichment of glycopeptides than the protein level-based protocol owing to lower-level contamination of non-glycosylated proteins or peptides during the enrichment process [49,50]. Currently, shaving *N*-glycan from Asn with PNGase F in heavy water (H_2_^18^O) to generate converted Asp with ^18^O labeling (+3 Da) is a feasible method in glycoproteomic analysis shown to enhance identifiable accuracy by eliminating the false glycosylated identification of chemical-inducible deamidation (−0.984 Da) during sample preparation [49,50,51,52]. In total, 10,572 peptides from AAL-affinity captured samples were identified. However, fewer peptides with ^18^O labeling (3.15%, 333/10,572) were acquired (Table 2), implicating significant non-specific retention of non-glycopeptides on the AAL affinity column prior to elution using 10% acetic acid (AA)/30% acetonitrile (ACN) in our study design. Replacement of elution buffer (10% AA/30% ACN) with PNGase F as a strategy for isotopic glycosidase elution and labeling on lectin column chromatography (IGEL), originally developed by the group of Ueda [49], may improve our design.

## 4. Materials and Methods

### 4.1. Plasma Samples

Peripheral blood samples were drawn from participants with standardized phlebotomy procedures and collected into an EDTA tube. After blood samples were centrifuged at 2000× *g*, plasma was isolated from the supernatants and then immediately aliquoted, transferred into plain polypropylene tubes, and stored in a dedicated freezer at −80 °C until use. A total of 121 plasma samples from patients were obtained from Chang Gung Memorial Hospital (Linkou, Taiwan) after informed consent had been acquired from all subjects. Of them, those with distant metastasis when the blood was drawn and tested for targeted molecules were designated as M1 (*n* = 44); while those without distant metastasis were designated as M0 (*n* = 27). Meanwhile, a cohort of 50 patients with gallstones (GS) subjected for elective cholecystectomy served as controls. Pathological staging of pancreatic cancer was based on AJCC edition 8. The detailed clinicopathological features are shown in Appendix A. The concentration of CA19-9 was measured by electrochemiluminescence immunoassay (ECLIA) with Roche cobas^®^ 8000 e602 analyzer (Roche Diagnostics, Rotkreuz, Switzerland).

### 4.2. Depletion of High-Abundance Plasma Proteins

Three plasma samples pooled from 10 GS controls, non-metastatic PC (M0) and metastatic PC (M1) patients were subjected to depletion of 14 highly abundant proteins using the Agilent Human 14 Multiple Affinity Removal System (MARS) column (4.6 × 100 mm; Agilent, Palo Alto, CA, USA), according to the procedure as previously described [53]. Briefly, each pooled sample (40 µL) was diluted 3-fold with 120 µL buffer A of the MARS column system. Diluted samples were processed with AKTA purifier 10 fast protein liquid chromatography (FPLC) (GE Healthcare Life Sciences, Piscataway, NJ, USA). Depleted fractions were desalted, concentrated, quantified and stored at −80 °C for further analysis.

### 4.3. Tryptic Digestion of Plasma Proteins and iTRAQ Labeling

Tryptic digestion of depleted plasma samples and iTRAQ labeling of digested peptides were performed as previously described [54]. Briefly, lyophilized depleted plasma samples (80 µg protein) were reduced, alkylated, and digested with 8 µg sequencing-grade modified porcine trypsin (1 µg/µL in trypsin resuspension buffer; Promega, Fitchburg, WI, USA) at 37 °C for 16 h. The resulting tryptic peptides of GS, M0, and M1 groups were labeled with iTRAQ tags 114, 115 and 116, respectively. Finally, iTRAQ-labeled products were pooled, separated into two fractions (30 µg and 200 µg) and desalted with solid-phase extraction Oasis HLB (30 µm) cartridges (Waters, Milford, MA, USA).

### 4.4. Glycopeptide Purification and Enzymatic Deglycosylation Integrated with ^18^O Labeling on Glycosylated Sites

Lyophilized, iTRAQ-labeled samples (200 µg peptide) were reconstituted with 1 mL lectin binding buffer (20 mM Tris, 0.3 M NaCl, 1 mM CaCl_2_, 1 mM MgCl_2_, pH 7.4) and incubated with 2 mg AAL agarose beads (Vector Laboratories, Burlingame, CA, USA) via rotation at room temperature for 1 h. After washing with lectin binding buffer three times and 500 µL ddH_2_O twice, AAL pull-down peptides were transferred to a new tube and eluted using 10% AA/30% ACN with shaking at room temperature for 10 min. Eluted peptides were re-lyophilized and dissolved in 50 mM sodium phosphate buffer, pH 7.5, supplemented with 90% H_2_^18^O and 40 U (1 unit/µL) *N*-glycosidase F (PNGase F; Roche Applied Science, Mannheim, Germany) and incubated at 37 °C with slight shaking for 20 h. Deglycosylated peptides were desalted with 40 µL C18 resin (source 15RPC, GE Healthcare, Björkgatan, Sweden) and lyophilized for further analysis using 2D-SCX/RP-LC-MS/MS.

### 4.5. Two-Dimension LC-MS/MS Analysis

Dried peptides including total plasma peptides (36 µg) and AAL-enrichment peptides were reconstituted in 30% ACN/0.1% formic acid (FA) and loaded onto home-made SCX column of two different sizes (0.5 × 200 mm and 0.5 × 100 mm; Luna SCX 5 µm, Phenomenex, Torrance, CA, USA) at flow rate of 3 µL/minute for 30 min. Peptides were eluted with 0–95% HPLC mobile phase (1 M ammonium nitrate/25% ACN/0.1% FA) and separated into 72 and 48 fractions using online 2D-HPLC (Dionex Ultimate 3000, Thermo Fisher, San Jose, CA, USA). Each SCX fraction was further 40-fold diluted in-line using 0.1% FA prior to trapping columns (Zorbax 300SB-C18 5 µm, 0.3 × 5 mm; Agilent Technologies, Wilmington, DE, USA) and diluted peptides resolved on an analytical C18 column (Synergi Hydro-RP 2.5 µm, 0.075 × 200 mm with a 15 µm tip; Phenomenex, Torrance, CA, USA), with 0–95% HPLC mobile phase (100% ACN/0.1% FA) at a flow rate of 0.25 µL/min. The LC apparatus was coupled to a two-dimensional linear ion trap mass spectrometer (LTQ-Orbitrap ELITE; Thermo Fisher, San Jose, CA, USA) controlled using Xcalibur 2.2 software (Thermo Fisher, San Jose, CA, USA), as previously described [54].

### 4.6. Mass Spectrometric Data Analysis

RAW files of resulting MS/MS spectra obtained from LTQ-Orbitrap MS were searched against the database containing 20,316 entries of *Homo sapiens* in SwissProt released on 4 March 2018 (https://www.uniprot.org/uniprot/?query=*&fil=reviewed%3Ayes+AND+organism%3A"Homo+sapiens+%28Human%29+%5B9606%5D" accessed on 31 March 2021) and commercial Proteome Discoverer 1.4 software (Thermo Fisher, San Jose, CA, USA) employed for data processing. The cleaved enzyme was set to “trypsin” with a maximum of one missed cleavage site. The precursor mass tolerance was set to 10 ppm and fragment ions mass tolerance to 0.5 Da for CID mode via ion trap analysis and 0.05 Da for HCD mode via Orbitrap analysis. The fixed modification was set to methylthiolation at cysteine (+45.99 Da) and variable modifications set to acetylation at protein N-terminus (+42.01 Da), oxidation at methionine (+15.99 Da), pyroglutamate conversion at N-terminal glutamine (−17.03 Da) and iTRAQ 4plex labeling at lysine and peptide N-terminal (+144.10 Da). Moreover, variable modifications of deamination at asparagine (N to D) and deamination with ^18^O labeling at asparagine (N to D_^18^O) were set for glycosylated site identification. Based on Mascot search results, the score threshold of peptide identification was set to “1% false discovery rate (FDR)” in the processing workflow and Peptide Validator algorithm was applied in calculation of FDR for peptide sequence analysis to distinguish true positives from random matches (decoy database). The decoy database was generated with Mascot and the size, including the number of amino acids and proteins, was the same as the original normal database [55]. In iTRAQ quantification, each reporter ion was integrated by the mode of most confident centroid at 20 ppm tolerance and iTRAQplex of 114 (114.11 Da) set as a denominator to compare other iTRAQplex of 115 (115.11 Da), 116 (116.11 Da) and 117 (117.11 Da) to generate a quantifiable ratio. Glycopeptides were processed using Byonic software (v2.12.0) (Protein Metrics Inc., Cupertino, CA, USA) in RAW files obtained from PC proteome analysis. The digested enzyme was set to “trypsin” with a maximum of one missed cleavage site. Mass tolerance and modification were the same as the above values. The library of 390 mammalian *N*-glycans was additionally set to the modified list for searching *N-*glycosylated peptides. Manual inspection was applied to confirm the confidence of identified glycopeptides and detect mass peaks of oxonium ion and glycan loss using Symbol Nomenclature for Glycans (SNFG). The mass spectrometry proteomics data were deposited to the ProteomeXchange Consortium via the PRIDE [56] partner repository (https://www.ebi.ac.uk/pride/login; Username: reviewer_pxd025150@ebi.ac.uk; Password: tE4tQjpe; Accessed date: 1 April 2021) with the dataset identifier PXD025150.

### 4.7. Reverse AAL-Based ELISA

AAL (100 µL; Vector Laboratories, Burlingame, CA, USA) (2.5 µg/mL, diluted in PBS buffer, pH 7.4) was added to wells of a 96-well plate (Corning Incorporated, Corning, NY, USA) and incubated at 4 °C overnight. After removal of unbound AAL, the plate was rinsed with washing buffer (0.1% Tween-20 in PBS, pH 7.4) (300 µL/well) six times and blocked with blocking buffer (T-Pro Biotechnology, New Taipei County, Taiwan) (150 µL/well) at room temperature for 2 h. Next, the plate was rinsed with washing buffer (350 µL/well) four times, and 100 µL serially diluted recombinant SERPINA1 protein (Research and Diagnostic Systems, Minneapolis, MN, USA) or plasma samples with 5-fold dilution (3% BSA/PBS) added to the wells and incubated at room temperature for 1 h. The plate was rinsed with washing buffer (300 µL/well) six times, and 100 µL Serpin A1 antibodies (Research and Diagnostic Systems, Minneapolis, MN, USA) (1 µg/mL) diluted in blocking buffer) added to each well and incubated at room temperature for 1 h, followed by six rinses with washing buffer (300 µL/well). Horseradish peroxidase conjugated-secondary antibody (100 µL; anti-mouse IgG HRP; Jackson ImmunoResearch Laboratories, West Grove, PA, USA) (1:3000, diluted in blocking buffer) was added to each well and incubated at room temperature for 40 min, followed by rinsing with washing buffer (300 µL/well) six times. Finally, 100 µL NeA-Blue Tetramethylbenzidine (TMB) solution (Clinical Science Products, Mansfield, MA, USA) was added to each well and incubated at room temperature for 30 min. The reaction terminated with 50 µL 2 N H_2_SO_4_ solution. Reaction products were measured with SpectraMax M5 Microplate Reader (MDS Inc., Toronto, ON, Canada) at an absorbance wavelength of 450 nm.

### 4.8. ELISA for SERPINA1 Protein

Plasma levels of SERPINA1 protein were measured using a commercially available ELISA kit (Sino Biological Inc., Beijing, China). Briefly, 100 µL mouse anti-SERPINA1 monoclonal antibodies, serving as capture antibodies (1 µg/mL, diluted in PBS buffer, pH 7.4), were coated onto a 96-well plate (Corning Incorporated, Corning, NY, USA) and incubated at 4 °C overnight. The plate was patted dry, rinsed with washing buffer (20 mM Tris, 150 mM NaCl and 0.05 % Tween-20, pH 7.2–7.4) (300 µL/well) three times, and blocked with blocking buffer (2% BSA in washing buffer) (300 µL/well) for 2 h. After rinsing the plate with washing buffer (350 µL/well) three times, 100 µL serially diluted recombinant SERPINA1 protein (as a standard curve) or plasma samples with 50,000-fold dilution in 0.1% BSA/washing buffer were added to wells and incubated at room temperature for 2 h. The plate was rinsed with washing buffer (300 µL/well) six times, and 100 µL rabbit anti-SERPINA1 monoclonal antibodies conjugated to horseradish peroxidase (detection antibodies; 0.4 µL/mL, diluted in 0.5% BSA/washing buffer) added to each well and incubated at room temperature for 1 h, followed by six rinses with washing buffer (300 µL/well). Finally, 100 µL TMB solution was added to each well and incubated for 20 min, and the reaction terminated with 50 µL 2 N H_2_SO_4_ solution. Reaction products were measured using a SpectraMax M5 Microplate Reader at an absorbance wavelength of 450 nm.

### 4.9. Statistical Analysis

The Mann–Whitney test was used to compare the differences in plasma levels of fuco-SERPINA1 and SERPINA1 proteins between GS and PC groups. Mann–Whitney and Kruskal–Wallis tests were used to evaluate the association of plasma fuco-SERPINA1, SERPINA1 and CA19-9 with various clinicopathological parameters of PC patients. Overall survival (OS) analysis was performed using the Kaplan–Meier method and differences in OS assessed via log-rank test. The diagnostic power of fuco-SERPINA1, SERPINA1 and CA19-9 was analyzed by constructing a receiver operating characteristic (ROC) curve with sensitivity versus 1-specificity and calculating area under the ROC curve (AUC). For all statistical analyses, a two-tailed *p*-value ≤ 0.05 was considered significant. Calculations and diagrams were generated using GraphPad Prism 7.0 (GraphPad Software, Inc., San Diego, CA, USA).

## 5. Conclusions

Our results demonstrated significant elevation of plasma fuco-SERPINA1 in PC patients compared to GS subjects, which was significantly associated with TNM stage and poor prognosis. Plasma fuco-SERPINA1 may therefore serve as a novel prognosticator for PC, which should be further examined on a larger scale in the future. The differentially expressed glycopeptides with definite glycosylation sites identified in this study present a valuable reservoir to explore novel biomarkers for PC.

## Figures and Tables

**Figure 1 ijms-22-06079-f001:**
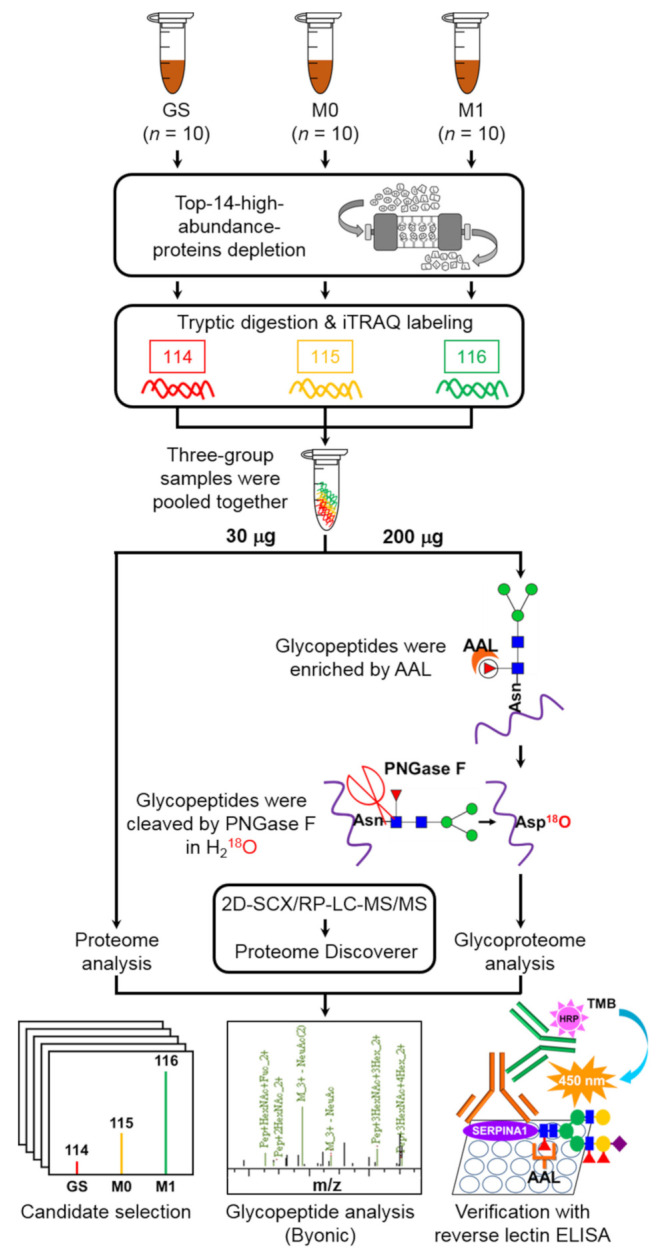
Schematic representation of the experimental design of this study. iTRAQ labeling coupled with 2D-SCX/RP-LC-MS/MS was applied for comprehensive analysis of the proteome profile of plasma samples from patients with PC, including non-metastatic (M0) and metastatic (M1) pancreatic cancers and subjects with gallstone lesions (GS). AAL was employed to enrich glycopeptides with specific glycan and H_2_^18^O to label glycosylated sites via PNGase F-mediated reaction for glycoproteome profiling analysis. We selected ^18^O-labeled fucose-containing glycopeptides upregulated in PC plasma samples as candidate targets. Fuco-SERPINA1, one of the candidate proteins, was selected for validation in plasma samples of PC and GS using reverse AAL-based ELISA.

**Figure 2 ijms-22-06079-f002:**
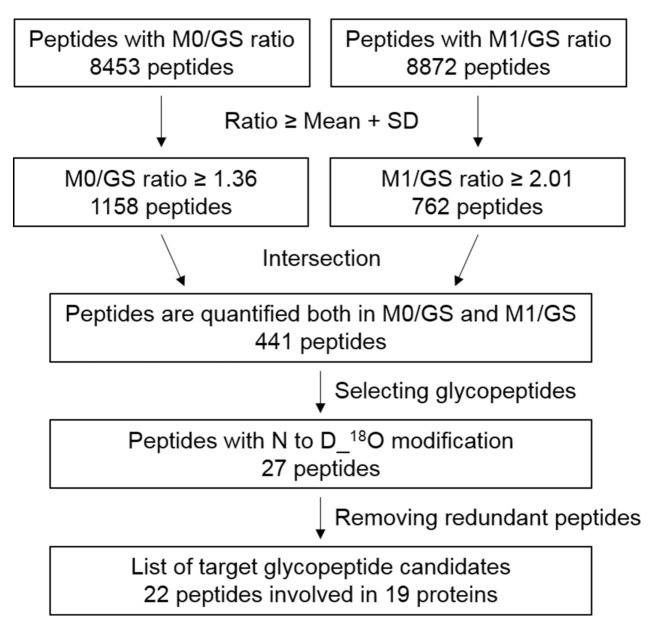
Flowchart for selecting candidate glycopeptides from glycoproteome analysis. From glycoproteome analysis, 8453 and 8872 peptides with quantified ratios in M0/GS and M1/GS groups were subjected to statistical analysis to determine mean and SD values. Peptides with ratios smaller than mean + SD were initially filtered out. The remaining 1158 and 762 peptides were intersected, leading to the identification of 441 peptides from which 27 harboring the modification of N to D_^18^O were further selected. After removing redundant peptides, 22 peptides representing 19 proteins were selected as target candidates.

**Figure 3 ijms-22-06079-f003:**
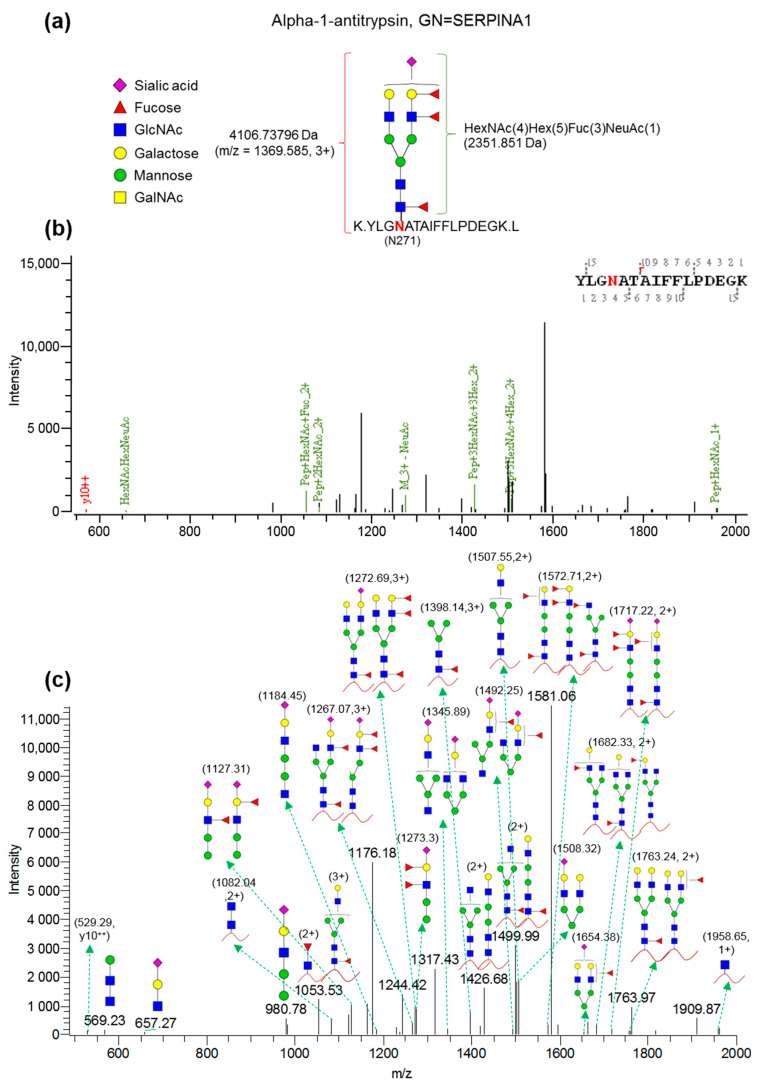
Analysis of the glycan structure of SERPINA1 (YLGNATAIFFLPDEGK). (**a**) The glycan structure of the target glycopeptide, YLGNATAIFFLPDEGK, identified in SERPINA1, was deciphered using Byonic software. (**b**) MS/MS spectrum of the targeted glycopeptide was analyzed and annotated using Byonic software. (**c**) Structures of glycan and glycopeptide were manually annotated in the MS/MS (CID) spectrum based on Byonic analysis.

**Figure 4 ijms-22-06079-f004:**
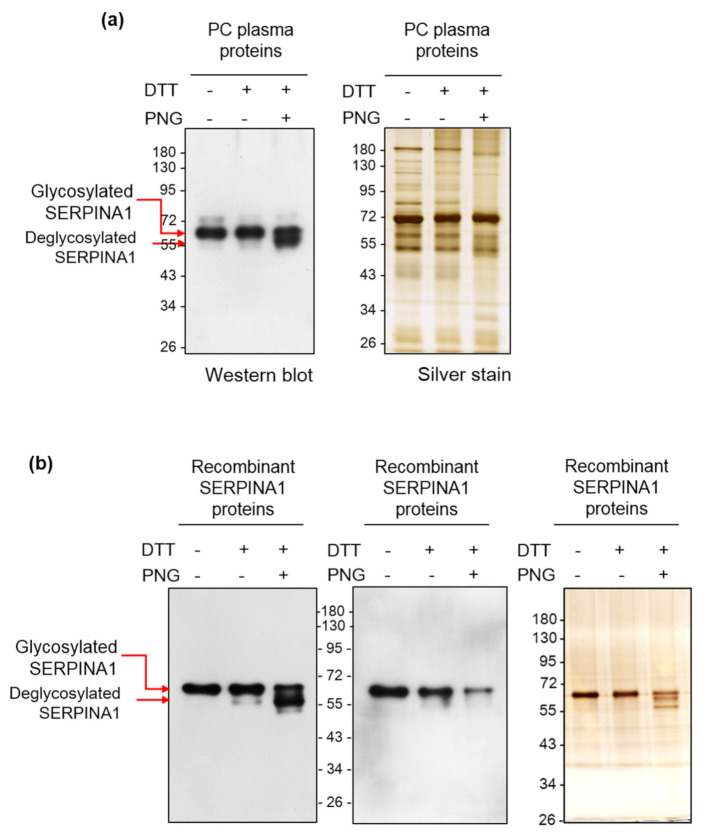
Deglycosylation of SERPINA1 protein by PNGase F alters its apparent molecular weight and AAL lectin-binding activity. (**a**,**b**) PC plasma proteins (2 µg) and recombinant SERPINA1 protein (200 ng) were treated with or without PNGase F (PNG) (protein/PNGase F = 10 µg/1U and 1 µg/1U, respectively) at 37 °C for 20 h and subjected to Western blot (for SERPINA1) or AAL blot analysis. Protein pattern revealed by silver staining is shown as loading control.

**Figure 5 ijms-22-06079-f005:**
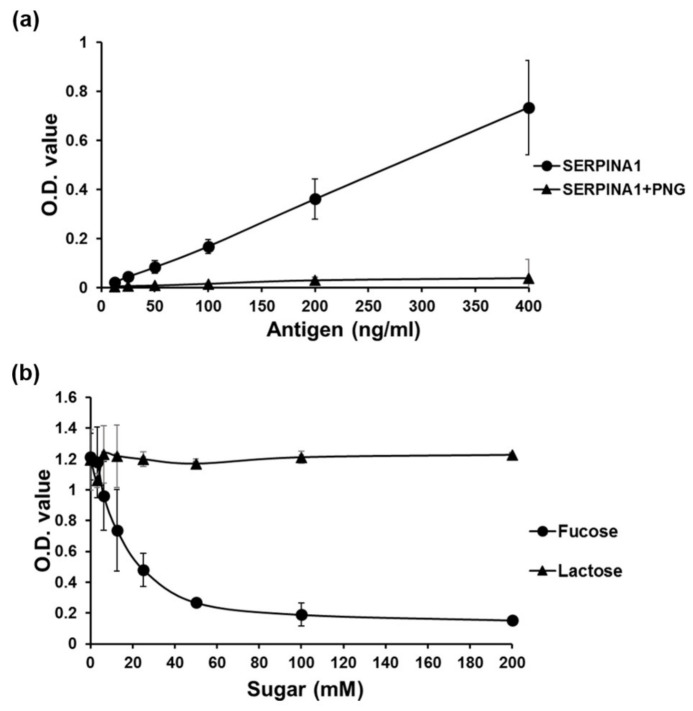
Evaluation of the specificity of AAL binding to fuco-SERPINA1 in reverse lectin ELISA. (**a**) Different amounts of recombinant SERPINA1 protein were treated with or without PNGase F (PNG) (protein:PNGase F = 1 µg:1U, 37 °C for 20 h) and subjected to AAL-based reverse lectin ELISA. (**b**) AAL was pre-incubated with different doses of L-fucose or lactose at room temperature for 0.5 h and coated onto ELISA plates. Plates were subjected to reverse lectin ELISA for detecting fixed amounts of fuco-SERPINA1. OD values were measured at 450 nm and each data point was examined in duplicate. Data expressed as mean ± S.D. are presented as a black solid line.

**Figure 6 ijms-22-06079-f006:**
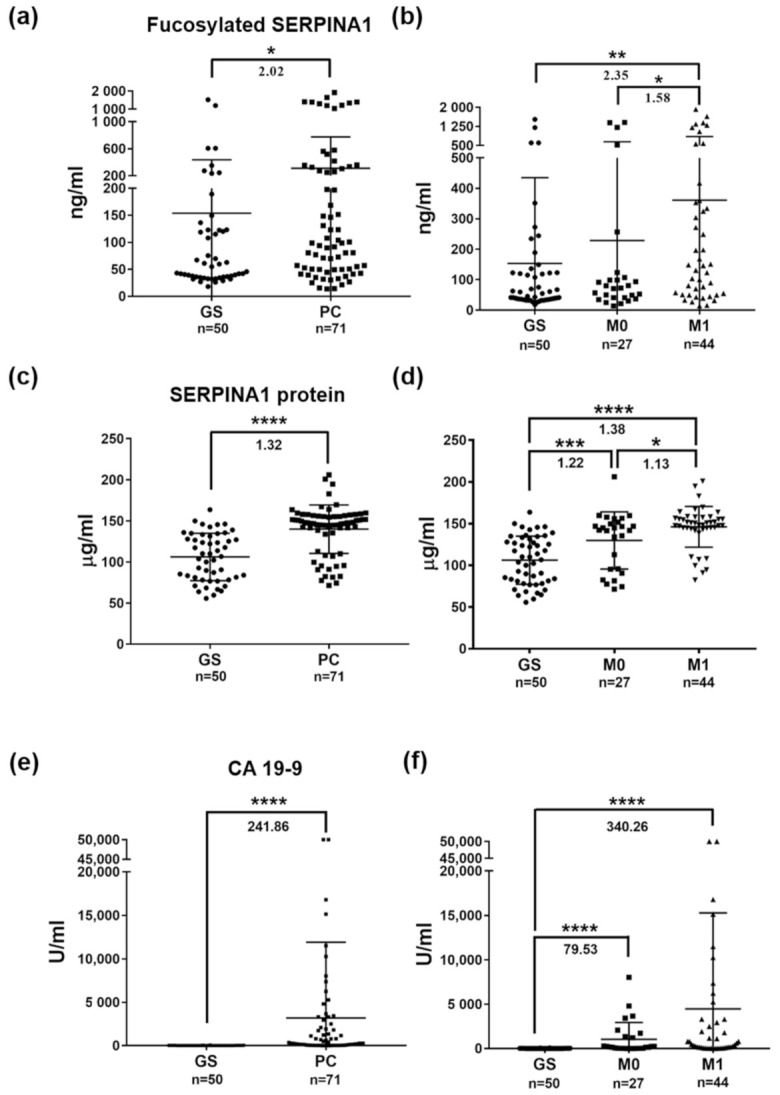
Plasma levels of fuco-SERPINA1, SERPINA1 and CA19-9 measured in enrolled subjects. (**a**,**b**) Levels of fuco-SERPINA1 measured in plasma specimens from GS and PC patients using AAL-based reverse lectin ELISA. (**c**,**d**) Levels of SERPINA1 protein measured in plasma specimens from GS and PC patients using commercial ELISA kits. (**e**,**f**) Levels of CA19-9 measured in plasma specimens from GS and PC patients using ECLIA. The horizontal lines indicate mean ± S.D. *, *p* ≤ 0.05; **, *p* ≤ 0.01; ***, *p* ≤ 0.001; ****, *p* ≤ 0.0001.

**Figure 7 ijms-22-06079-f007:**
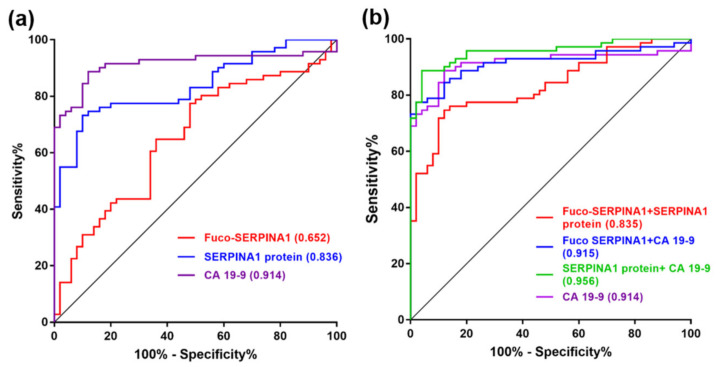
ROC curve analysis of fuco-SERPINA1, SERPINA1 protein and CA19-9 in discriminating PC patients from GS subjects. (**a**) The discriminatory power of plasma fuco-SERPINA1, SERPINA1 and CA19-9 was evaluated via ROC curve analysis. (**b**) The discriminatory power of combined fuco-SERPINA1 with SERPINA1 or CA19-9 was further evaluated. AUC values are presented in brackets.

**Figure 8 ijms-22-06079-f008:**
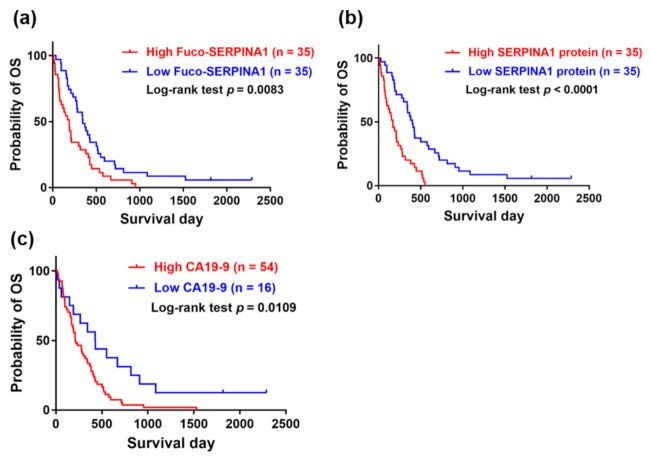
Association between overall survival of PC patients and plasma levels of fuco-SERPINA1 (**a**), SERPINA1 (**b**) and CA19-9 (**c**) analyzed using Kaplan–Meier plot. The time of survival was assessed in 68 of 71 PC patients enrolled in this study. The log-rank test *p*-value is denoted in each plot.

**Table 1 ijms-22-06079-t001:** Clinicopathological characteristics of the enrolled subjects used in this study.

Characteristics		Gallstones (GS)	Non-Metastatic PC (M0)	Metastatic PC (M1)
(For Discovery Experiment, *n* = 30)		-	-	-
Gender	Female	6	5	1
	Male	4	5	9
Age (years) *^a^*		62.9 ± 11.0	61.5 ± 9.8	59.9 ± 9.3
Tumor size (T)	T3	-	4	4
	T4	-	6	6
Lymph node metastasis (N)	Yes	-	8	8
	No	-	2	2
Distant metastasis (M)	Yes	-	0	10
	No	-	10	0
Stage	I–II	-	4	0
	III–IV	-	6	10
**(Total enrolled subjects, *n* = 121)**		-	-	-
Gender	Female	29	12	12
	Male	21	15	32
Age (years)		54.4 ± 13.2	61.3 ± 12.2	62.4 ± 9.9
Tumor size (T)	T1	-	1	0
	T2	-	1	4
	T3	-	15	23
	T4	-	10	16
Lymph node metastasis (N)	Yes	-	21	37
	No	-	6	7
Distant metastasis (M)	Yes	-	0	44
	No	-	27	0
Stage	I–II	-	17	0
	III–IV	-	10	44

*^a^* Data are shown in mean ± standard deviation (SD).

**Table 2 ijms-22-06079-t002:** Numbers of identified proteins, peptides and glycopeptides in plasma proteome and glycoproteome.

Identified Proteins, Peptides and Glycopeptides	Quantitative Proteome Profiling			Quantitative Glycoproteome Profiling		
	GS + M0 + M1	M0/GS	M1/GS	GS + M0 + M1	M0/GS	M1/GS
Total proteins	1707	1489	1472	2160	1749	1953
N to D_^18^O proteins	-	-	-	145	133	130
Total peptides	10,102	8621	8304	10,572	8453	8872
N to D_^18^O peptides	-	-	-	333	281	267
Ratio (N to D_^18^O peptides/total peptides)	-	-	-	3.15%	3.32%	3.01%
N to D_^18^O and NXS/T/C peptides	-	-	-	284	244	232
Ratio (N to D_^18^O and NXS/T/C peptides/N to D_^18^O peptides)	-	-	-	85.29%	86.83%	86.89%

N to D_^18^O, asparagine (N) was transferred to aspartate (D) by replacing -NH_2_ with -^18^OH via glycosidase treatment (PNGase F) in H_2_^18^O-containing buffer. NXS/T/C, consensus sequence of *N*-linkage glycosylation, where X represents any amino acid except proline.

**Table 4 ijms-22-06079-t004:** Correlations of plasma fucosylated SERPINA1, SERPINA1, and CA19-9 levels with clinicopathological characteristics of PC patients.

Characteristics	Number	Fucosylated SERPINA1 (ng/mL)	*p*-Value	SERPINA1 protein (µg/mL)	*p*-Value	CA19-9 (U/mL)	*p*-Value
Gender *^a^*	-	-	-	-	-	-	-
Male	47	332.6 ± 482.1	0.949	142.1 ± 28.5	0.450	3329 ± 10,192	0.693
Female	24	267.8 ± 438.8	-	135.6 ± 31.4	-	2854 ± 4884	-
Age (years) *^a^* <62 *^c^*	- 35	- 302.5 ± 422.2	- 0.862	- 140.8 ± 29.8	- 0.905	- 4572 ± 11,843	- 0.844
≥62	36	318.7 ± 509.3	-	139 ± 29.5	-	1804 ± 3488	-
TNM stage *^b^* Stage I and II	- 17	- 201.1.7 ± 418.4	- 0.024 *^d^*	- 130.2 ± 36.5	- 0.060	- 781.1 ± 1191	- 0.222
Stage III	10	275.3 ± 423.2	-	129 ± 31.7	-	1495 ± 2753	-
Stage IV	44	361.1 ± 491.8	-	146.1 ± 24.5	-	4471 ± 10,827	-
Tumor stage *^b^* T1 and T2	- 6	- 267.0 ± 508.0	- 0.300	- 137.2 ± 32.6	- 0.774	- 319 ± 371	- 0.336
T3	38	281.3 ± 444.2	-	141.7 ± 30.7	-	4516 ± 11,447	-
T4	26	301.8 ± 395.7	-	137.8 ± 28.4	-	1976 ± 3565	-
Lymph node metastasis *^a^* N0	- 13	- 327.2 ± 573.2	- 0.610	- 136.4 ± 36.9	- 0.994	- 5447 ± 13,704	- 0.799
N1	58	307 ± 443.2	-	140.7 ± 27.8	-	2658 ± 7248	-
Distant metastasis *^a^* M0	- 27	- 228.6 ± 413.6	- 0.043 *^d^*	- 129.7 ± 34.2	- 0.017 *^d^*	- 1045 ± 1903	- 0.097
M1	44	361.1 ± 491.8	-	146.1 ± 24.5	-	4471 ± 10,827	-

*p*-values were evaluated using *^a^* Mann–Whitney test (mean ± SD) or *^b^* Kruskal–Wallis test (mean ± SD). *^c^* Threshold of age was determined based on the median of patient ages. *^d^* Statistically significant, *p*-value ≤ 0.05.

## Data Availability

The data that support the findings of this study are available from the corresponding author upon reasonable request.

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
