# Peer review of "Identification of Fucosylated SERPINA1 as a Novel Plasma Marker for Pancreatic Cancer Using Lectin Affinity Capture Coupled with iTRAQ-Based Quantitative Glycoproteomics"

_ijms, 2021, doi:10.3390/ijms22116079_

Round 1
Reviewer 1 Report
The manuscript entitled “Identification of Fucosylated SERPINA1 as a Novel Plasma Marker for Pancreatic Cancer using Lectin Affinity Capture Coupled with iTRAQ-Based Quantitative Glycoproteomics” by Wu and colleagues identified fuco-SERPINA1 levels as a novel circulating prognostic biomarker for Pancreatic cancer. The study has been conducted on a cohort of n=20 pancreatic patients and n=10 subjects with gallstones, as control group. Overall, this topic is quite important and interesting. This work can help to better understand the role of SERPINA1 as prognostic marker in Pancreatic Cancer. In my opinion, it can be accepted after minor revision.
Limitations.
Results and methods should be more concise
Strengths
Novel and interesting findings presented
Experimental design correctly performed
I have only few comments
- The rationale behind the selection of SERPINA1 among other identified genes for ELISA analyses should be included in the aim.
- Describing methodologies in the results section makes this section difficult to follow. Where possible, I suggest to move these sentences to the methods section
- As correctly stated in the discussion, SERPINA1/ Alpha-1 anti-trypsin (AAT) protein levels increase during inflammation and in several diseases. However, the levels of this protein have also been found to rise in normal physiological conditions, such as pregnancy (PMID: 33015055). This is an important information that should be included in the discussion section, along with the supporting reference.
- SERPINA1 is an inducible gene mainly expressed in hepatocytes, monocytes and macrophages. It encodes for AAT a serine protease inhibitor mainly synthesized by the liver, and released into the bloodstream (PMID: 22365503). This information and supporting reference should be included in the discussion.
- Methods are long and should be presented more concisely. I would be helpful for the reader
Minor comments
- Line 90-> Please remove the additional parenthesis
- Discussion, AACT, A1AT, LRG, THBS1 and others should be mentioned as complete name when quoted the first time
Reviewer 2 Report
In present manuscript authors have demonstrated that fucosylated SERPINA1 protein level in plasma of the pancreatic cancer patients plasma may serve as prognostic marker, although large scale validation studies needed. By utilizing iTRAQ quantitative proteomics approach along with AAL based glycopeptide enrichment and isotope-coded glycosylation site-specific tagging, they analyzed the plasma glycoprotein of metastatic, non-metastatic pancreatic cancer and gallstone patients. Their result demonstrates that fucosylated SERPINA1 level was significantly high in pancreatic cancer patents plasma in comparison to galstone patients. Increased level of fuco-SERPINA1was correlated with TNM stage. Findings are very important in order to develop the prognostic marker. I have following concerns regarding the study.
1- Except SERPINA1, does any other fucosylated proteins which authors identified in their analysis, have prognostic significance or have been reported previously in ant type of cancer?
2-Have authors evaluated any other modification of SERPINA1 such as phosphorylation/acetylation etc. as control? It should be evaluated to establish fucosylation specific prognostic significance of SERPINA1.
3-Loading control for immunoblots needs to be provided.
4- Figure 3B, axis labels needs to be improved.
5-Were there any glycopeptides were significantly different between metastatic vs non-metastatic patients plasma? Authors need to discuss these findings.
